# Universal Ordering for Efficient PAC Learning

## Abstract

We initiate the study of the *universal ordering* problem within the PAC learning framework: given a set of $n$ samples independently drawn from an unknown distribution $\mathcal{D}$, can we order these samples such that every prefix of length $k \leq n$ yields a near-optimal subset for training a PAC learner? This question is fundamentally motivated by practical scenarios involving incremental learning and adaptive computation, where guarantees must hold uniformly across varying data budgets. We formalize this requirement as achieving anytime-valid PAC guarantees. As a warm-up, we analyze the simple random ordering baseline using classical concentration inequalities. Through a careful union bound over a geometric partitioning of prefixes, we establish that it provides a surprisingly strong universal guarantee, incurring at most an $O(\log \log n)$ overhead compared to a random subset of size $k$. We then present a more powerful analysis based on the theory of test martingales and Ville's inequality, demonstrating that a random permutation achieves PAC guarantees for all prefixes that match the statistical rate of a random subset of size $k$, without the logarithmic overhead incurred by naive union-bound techniques. Our work establishes a conceptual bridge between universal learning on fixed datasets and the broader field of sequential analysis, revealing that random permutations are efficient and provably robust anytime-valid learners but opening the door to further improvements.

## 1 Introduction

Modern machine learning increasingly deals with massive datasets that significantly exceed practical computational capacities, rendering it infeasible to utilize all available data simultaneously (Bachem et al., 2017; Kettenring, 2009; Labrinidis & Jagadish, 2012). Consequently, practitioners commonly resort to selecting representative subsets of data to train algorithms effectively within stringent computational budgets (Bubeck et al., 2019; Muennighoff et al., 2023; Thompson et al., 2020). Classical Probably Approximately Correct (PAC) learning theory provides fundamental guarantees regarding the minimum number of samples necessary to achieve prescribed accuracy and confidence thresholds (Kearns & Vazirani, 1994). However, traditional PAC bounds assume a static, predetermined sample size. In sharp contrast, practical scenarios frequently involve dynamic computational budgets, requiring robust guarantees that hold simultaneously across multiple scales of data usage (Jiang et al., 2020; McIntosh et al., 2018).

Addressing this critical gap motivates our systematic investigation of the **universal ordering problem**: given $n$ samples drawn i.i.d. from an unknown distribution $\mathcal{D}$, can these samples be arranged in a fixed sequence such that every initial segment (or prefix) of length $k \leq n$ forms an approximately optimal subset for PAC learning? We formalize this desideratum through the notion of Universal PAC-Validity. This requirement is structurally identical to the demand for a confidence sequence in modern sequential analysis—a sequence of confidence intervals that are guaranteed to contain the true parameter of interest uniformly over all time steps (Waudby-Smith & Ramdas, 2020). In our context, the "time step" is the prefix length $k$, and the parameter of interest is the true error of the hypothesis learned from that prefix. This reframing is not merely semantic; it allows for the deployment of powerful tools from sequential analysis that are designed to handle such uniform-over-time guarantees.

The universal ordering problem holds considerable practical relevance. Consider the crucial goal of reproducibility and fair benchmarking in machine learning. A fixed, universal ordering for a benchmark dataset (e.g., ImageNet (Deng et al., 2009)) would ensure that researchers comparing

models with different computational budgets are all training on valid, nested subsets of the same data sequence. This allows us to view the learning process as traversing a valid confidence sequence: a model trained on a prefix of size 100,000 is directly comparable to one trained on the first 1,000,000 points, as both are certified snapshots along the same statistical trajectory. Furthermore, in scenarios of resource-adaptive learning, a model may train on a device with a variable power budget or on a shared cluster where it can be preempted at any time. A universal ordering ensures that if the process halts at an arbitrary point $k$, the resulting model is not just a partial result but one that comes with a valid PAC guarantee for the data seen so far.

While related concepts such as coreset (Chai et al., 2023), curriculum learning (Bengio et al., 2009), and submodular optimization (Mirzasoleiman et al., 2013) have been extensively explored, these existing methodologies typically target subset construction tailored to a predetermined size or employ heuristic-based approaches that lack robust guarantees for dynamically varying data sizes. Consequently, the universal ordering problem delineates a new and compelling intersection among combinatorial optimization, statistical learning theory, and adaptive computational frameworks.

The primary difficulty arises from the "for all $k$" quantifier in the problem definition. From a classical statistical perspective, this introduces a severe multiple testing problem. A naive analysis using standard concentration inequalities would require applying a union bound over all $n$ prefixes, incurring a substantial statistical penalty that would render the resulting error bounds vacuous. This challenge underscores the need for more sophisticated analytical techniques that can account for the strong dependencies between hypotheses trained on nested prefixes.

## 1.1 OUR CONTRIBUTIONS

This paper provides a comprehensive theoretical analysis of random permutations as a first solution to the universal ordering problem as a means to present two distinct but complementary analytical frameworks, probing the noted multiple testing problem.

1. First, we formalize the universal ordering problem and establish a strong baseline for task-agnostic random permutations using a classical analysis (and further discuss optimality under the task-agnostic constraint in Appendix A.3). This approach, based on a careful union bound over a geometric partitioning of prefixes, reveals that a random ordering incurs a surprisingly small $O(\log \log n)$ overhead in sample complexity compared to an optimal random subset selected for a specific size $k$. This result serves as a valuable warm-up and demonstrates the inherent robustness of random shuffling.

2. Second, we introduce a more direct and powerful analysis rooted in the theory of anytime-valid inference (Robbins & Siegmund, 1974; Wald, 2004). By constructing a specific test supermartingale for each potentially "bad" hypothesis, we leverage Ville's inequality (Wald, 2004) to provide a uniform guarantee over all prefixes. This approach is more elegant, avoids the need for explicit union bounds over prefixes, and yields a tighter bound that removes the logarithmic factors entirely. The construction of this martingale is informed by a key observation: a random permutation of a fixed dataset is equivalent to sampling without replacement from a finite population, allowing us to adapt powerful martingale constructions from that literature (Hall & Heyde, 2014).

3. Third, we establish a conceptual bridge between the universal ordering problem in PAC learning and the broader fields of sequential analysis and safe testing (Grünwald et al., 2024; Ramdas et al., 2023). This connection suggests that the principles of designing and analyzing data orderings for robust, incremental performance have wide applicability beyond the standard PAC framework.

Most crucially, by defining this problem and the strengths of different analytic approaches, we hope to inspire future work on improved (task optimal) data ordering approaches.

## 2 RELATED WORK

Our universal ordering problem intersects multiple domains within combinatorial optimization, machine and statistical learning theory (Shalev-Shwartz & Ben-David, 2014; Vapnik & Chervonenkis,

2015). Largely, our work is grounded in the tradition of PAC learning, which provides formal guarantees on a model's generalization performance (Kearns & Vazirani, 1994). While foundational, our work departs from the standard PAC setting by focusing on incremental performance guarantees across subsets of a single dataset rather than on learning a single hypothesis for one underlying distribution. Closely related extensions of the PAC learning framework to the present work, such as collaborative PAC learning (Blum et al., 2017), often consider scenarios involving multiple learners working collaboratively to find an optimal hypothesis across distinct distributions. However, unlike collaborative PAC learning, which emphasizes multi-distribution scenarios, our universal ordering framework focuses explicitly on incremental guarantees across subsets of a single dataset.

Our problem formulation is conceptually connected to classical problems in universal approximation algorithms and incremental optimization (Lin et al., 2010). For instance, universal approximations for the Steiner tree and set cover problems (Jia et al., 2005) aim to identify single solutions or structures that approximately solve combinatorial optimization problems simultaneously under multiple potential inputs or constraints. Similarly, oblivious network design (Gupta et al., 2006), the universal traveling salesman problem and related routing challenges (Jia et al., 2005; Schalekamp & Shmoys, 2008) explore scenarios that require performance guarantees across multiple, dynamically varying instances without prior knowledge of specific instance parameters. These works underscore the broader theoretical difficulty inherent in obtaining universal or incremental performance guarantees, highlighting the analytical challenges in the present problem context.

Additionally, extensive literature has examined sufficient summarization techniques through coresets and related subset selection methodologies for diverse learning and optimization problems (Mirzasoleiman et al., 2020; Bachem et al., 2017; Phillips, 2017). These approaches typically focus on constructing fixed-budget approximations for specific tasks. In contrast, our universal approach uniquely aims to identify a single sequence of points with simultaneous guarantees across all input subset sizes through a single computationally efficient pass.

Finally, curriculum learning (Bengio et al., 2009; Hacohen & Weinshall, 2019; Weinshall et al., 2018) offers empirically successful heuristics for ordering data to accelerate model convergence or enhance performance. This problem deviates from the present in two crucial ways: (1) curriculum learning sequentially presents data with the goal of achieving an improved model *at the end of the full data sequence* and (2) relies on expensive model fitting and data diagnostics for each sample to examine how it will contribute to the final model's performance. In contrast, we seek to compute a single, computationally efficient, pass over the data such that any subsequence of the returned ordered is nearly optimal. Despite its practical effectiveness, curriculum learning lacks the necessary uniform theoretical guarantees for such prefix lengths. Active learning, a related problem, involves iterative querying of an oracle to select data points sequentially, optimizing marginal information gains. Our problem instead assumes a fixed dataset without additional queries and must return a static ordering, rather than incrementally include query for points to incorporate into the training data.

## 3 PRELIMINARIES

We adopt the standard PAC learning framework (Kearns & Vazirani, 1994). Let $\mathcal{X}$ be a domain and $\mathcal{Y}$ be a label set. A hypothesis $h : \mathcal{X} \to \mathcal{Y}$ is drawn from a class $\mathcal{H}$. Given a distribution $\mathcal{D}$ over $\mathcal{X} \times \mathcal{Y}$, the population loss (or error) is $err_{\mathcal{D}}(h) = \mathbb{E}_{(x,y)\sim\mathcal{D}}[\mathbf{1}[h(x) \neq y]]$. A learning algorithm $\mathcal{A}$ maps a sample $S$ to a hypothesis $h \in \mathcal{H}$.

**Definition 3.1** (($\varepsilon, \delta$)-PAC Learnable). A hypothesis class $\mathcal{H}$ is PAC learnable if there exists an algorithm $\mathcal{A}$ and a function $n_{\mathcal{H}}(\varepsilon, \delta)$ such that for any distribution $\mathcal{D}$, given $n \geq n_{\mathcal{H}}(\varepsilon, \delta)$ i.i.d. samples, $\mathcal{A}$ returns a hypothesis $h$ satisfying $err_{\mathcal{D}}(h) \leq \varepsilon$ with probability at least $1 - \delta$.

Throughout our results, $\varepsilon$ is used to denote the error rate bound for a model trained on $n$ samples. For a finite hypothesis class, we have the following standard result on the sample complexity.

**Theorem 3.2** (Finite Hypothesis Class Sample Complexity (Kearns & Vazirani, 1994)). *Let $\mathcal{A}$ be an algorithm that learns a finite hypothesis class $\mathcal{H}$ in the consistency model (that is, returns $h \in \mathcal{H}$ whenever a consistent concept w.r.t. $S$ exists). Then, $\mathcal{A}$ learns the concept class $\mathcal{H}$ in the PAC*

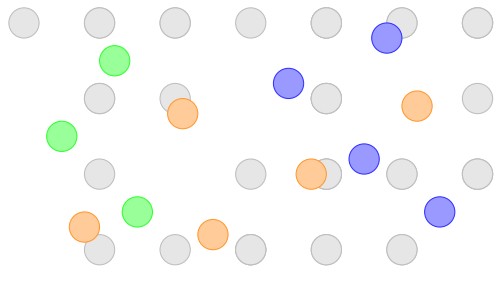
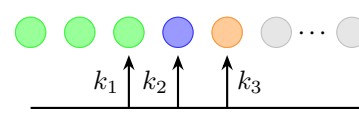

| (a) Subsets are not nested | (b) Subsets are inherently nested |

Figure 1: Conceptualizing independent random subsets vs. universal ordering. (a) shows traditional independent random subsets for different data budgets ($k_1 = 3, k_2 = 4, k_3 = 5$) which are generally not nested. (b) illustrates a universal ordering, where any prefix naturally forms a nested subset for a given budget, a property crucial for anytime-valid guarantees.

*learning model using*

$$n_{\varepsilon,\delta} \in \mathcal{O}\left(\frac{\log|\mathcal{H}| + \log\frac{1}{\delta}}{\varepsilon}\right).$$

For the broader class of hypotheses with finite VC dimension $d = \text{VC}(\mathcal{H})$, we further have the following bound.

**Theorem 3.3** (Infinite Hypothesis Class Sample Complexity (Hanneke, 2016)). *Let $\mathcal{A}$ be an algorithm that learns a hypothesis class $\mathcal{H}$ in the consistency model (that is, returns $h \in \mathcal{H}$ whenever a consistent concept w.r.t. $S$ exists). Then, $\mathcal{A}$ learns the concept class $\mathcal{F}$ in the PAC learning model using*

$$n_{\varepsilon,\delta} \in \Theta\left(\frac{1}{\varepsilon}\left(d + \log\frac{1}{\delta}\right)\right).$$

Throughout our analysis for infinite classes, we assume the prefix length $k$ is at least $d$, as guarantees are not meaningful otherwise.

### 3.1 UNIVERSAL PAC-VALIDITY CRITERION

We now formalize the central notion of our work.

**Definition 3.4** (Universal PAC-Validity). Let $S_n = (z_1, \ldots, z_n)$ be a sequence of $n$ examples drawn from a distribution $\mathcal{D}$. We say that $S_n$ is universally PAC-valid for a learner $\mathcal{A}$ with error bound sequence $\{\varepsilon_k\}_{k=1}^n$ and confidence $1 - \delta$ if, with probability at least $1 - \delta$ over the generation of $S_n$, the sequence of hypotheses $h_k = \mathcal{A}(S_k)$ satisfies $err_{\mathcal{D}}(h_k) \leq \varepsilon_k$ for all $k \in [n]$.

This definition requires a single sequence to support correct generalization across all prefixes, which, as noted, is equivalent to constructing a confidence sequence for the true error of the learner at each prefix length (Waudby-Smith & Ramdas, 2020).

We here briefly note the discrepancy between the universal bounds we explore and simpler notions of PAC complexity on samples of size $k \leq n$. Observe that if we randomly sample $k$ points from the distribution $\mathcal{D}$, we can trivially apply the result of Theorem 3.2 to obtain an error bound of at most $\mathcal{O}\left(n\varepsilon/k\right)$ where $\varepsilon$ is the error rate on $n$ samples. However, this error bound holds with probability $1 - \delta$ for only this value of $k$. In order to obtain a bound which holds for all $k$ at the same error rate, we must naturally degrade our error bound and refer to the additional error incurred (as a multiplicative factor to $\varepsilon$) as the *overhead*.

### 3.2 MARTINGALES AND VILLE'S INEQUALITY FOR SEQUENTIAL GUARANTEES

Our main results rely on the theory of martingales, which provides a principled framework for analyzing sequential processes.

**Definition 3.5.** A sequence of random variables $(M_k)_{k \geq 0}$ is a supermartingale with respect to a filtration $(\mathcal{F}_k)_{k \geq 0}$ (an increasing sequence of $\sigma$-algebras representing information available at time $k$) if for all $k \geq 0$:

1. $M_k$ is $\mathcal{F}_k$-measurable.

2. $\mathbb{E}[|M_k|] < \infty$.

3. $\mathbb{E}[M_{k+1}|\mathcal{F}_k] \leq M_k$.

A non-negative supermartingale is a powerful tool for deriving concentration inequalities. The following result, Ville's inequality, is a time-uniform extension of Markov's inequality and forms the mathematical engine of our improved analysis.

**Theorem 3.6** (Ville's Inequality (Wald, 2004))**.** *Let $(M_k)_{k \geq 0}$ be a non-negative supermartingale with $M_0 \leq 1$. Then for any $\alpha \in (0, 1)$:*

$$P\left(\exists k \geq 0 : M_k \geq \frac{1}{\alpha}\right) \leq \alpha$$

Ville's inequality converts a statement about one-step-ahead expectations into a strong probabilistic bound on the entire trajectory of the process (Shafer & Vovk, 2019). This allows us to control the "bad events" over all prefixes $k$ simultaneously without incurring the penalty of a union bound. The intuition behind the "stability of consistency" lemma, which we discuss in Section 4.2, is captured formally by this supermartingale property.

## 4   WARM-UP: CLASSICAL UNION BOUND ANALYSIS

We first the universal guarantees that come from a standard union bounding argument, highlighting the deficiency in this method and motivating our later study of anytime-valid approaches.

### 4.1   A NAIVE LOGARITHMIC BOUND

For a finite hypothesis class $\mathcal{H}$, we can bound the probability of a "bad event" at a fixed prefix $k$ (i.e., a consistent hypothesis having high error) and sum these probabilities.

**Theorem 4.1.** *Let $\mathcal{H}$ be a finite hypothesis class and $\mathcal{A}$ a consistent learning algorithm. A random order of $n$ examples $S_n = (z_1, ..., z_n)$ drawn i.i.d. from $\mathcal{D}$ is universally PAC-valid with error at most $\min\{\frac{n\varepsilon + \log n}{k}, 1\}$ and confidence $1 - \delta$, provided $n$ is large enough for $(\varepsilon, \delta)$-PAC learnability.*

*Proof Sketch.* We proceed by considering a fixed prefix length, $k$, and bound the probability of the bad event that the corresponding hypothesis, $h_k$, has large despite being consistent with the prefix. More formally, we bound the probability that $h_k$ is consistent given that its error is at least $\varepsilon_k$. This probability is equivalent to a Bernoulli trial and can be upper bounded as $(1 - \varepsilon_k)^k \leq e^{-k\varepsilon_k}$. Taking a union bound over the the hypothesis hypothesis class, we obtain a bound for the failure probability at a fixed $k$ value. To ensure the at most $n$ prefixes satisfy the desired error guarantee of $\varepsilon$ corresponding to the sample complexity $n_{\varepsilon, \delta}$, we apply a union bound over all the event failure probablities to obtain the adaptive bound $\varepsilon_k \geq \frac{n\varepsilon + \log n}{k}$. The full proof is detailed in Appendix A. $\square$

Thus, the overhead for ensuring universality is at most logarithmic in the overall sample complexity. If we were to instead select $k$ data points at random, the standard PAC learning results would guarantee an error of at most $\frac{n\varepsilon}{k}$ with probability $1 - \delta$. However, we reiterate that our framing seeks to define a bound on the $k$-sized prefix training set which holds with high probability across all such values of $k$, incurring an additional logarithmic error. In the next section, we show how this bound can be tightened significantly by recognizing that adjacent prefixes are highly correlated, allowing us to control far fewer "bad events", improving the overhead to $O(\log \log n)$.

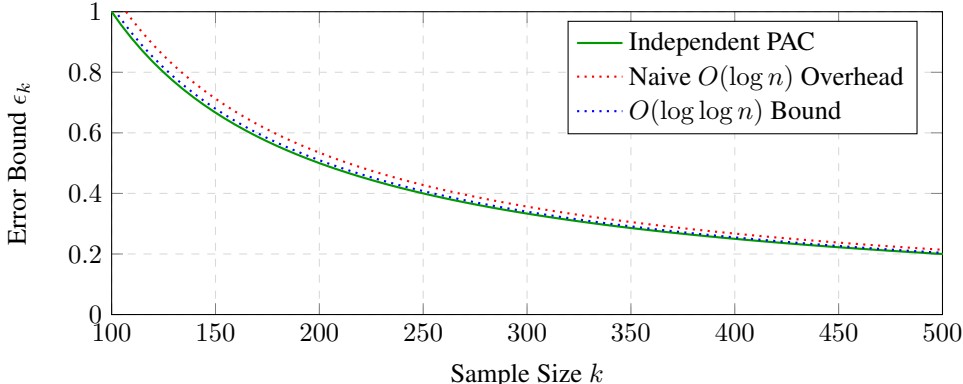

Figure 2: Comparison of Generalization Error Bounds. Naive union bound analysis (dotted red) incurs pessimistic logarithmic overhead, whereas a double counting argument yields an exponential improvement dotted blue) that nearly matches independent random sampling.

## 4.2 An Exponential Improvement to a $O(\log \log n)$ Overhead

The prior analysis assumed a conservative worst-case scenario where each prefix length was an independent event, requiring us to control the generalization error separately for all $n$ prefixes. However, a crucial structural property of data orderings allows us to significantly tighten this analysis. The key insight is that hypotheses trained on similar-sized prefixes are highly correlated. Specifically, if a hypothesis is consistent on a prefix of size $k$, it is likely to remain consistent on slightly longer prefixes, e.g. those of size $(1+\eta)k$ for small $\eta > 0$. This suggests that controlling error at exponentially spaced prefix sizes suffices to ensure correctness for all intermediate values.

Formally, we construct a geometrically growing sequence of prefix sizes and bound the error only at these anchor points. Because consistency at size $k$ propagates to $(1 + \eta)k$ with high probability, we need only apply a union bound over $O(\log n)$ such anchor prefixes. This reduces the overhead to $\log \log n$. We note that this reduction is not merely a technical improvement, but rather it underscores a key qualitative advantage of universal orderings: small-scale generalization guarantees can be leveraged (and propagated) to ensure performance at much larger scales. We proceed to formalize this insight.

**Lemma 4.2** (Stability of Consistency under Prefix Extension). *Let $h_k$ be a hypothesis consistent with the first $k$ samples of a random ordering $S_n = (z_1, ..., z_n)$ drawn i.i.d. from a distribution $\mathcal{D}$, and assume $err_{\mathcal{D}}(h_k) \leq \epsilon_k$. Then for any $\eta \in (0, 1]$, the probability that $h_k$ is also consistent with the next $\eta k$ examples in the sequence is at least $(1 - \epsilon_k)^{\eta k}$.*

This fact applies to both finite and infinite hypothesis class settings, as it relies solely on the generalization error bound of the hypothesis $h_k$, rather than the size of the class itself. As a result, it suffices to control the generalization error only at exponentially spaced prefixes of the form $k_j = (1+\eta)^j$, reducing the total number of bad events from $n$ to $O(\log n)$. We proceed to revise the proof of Theorem 4.1 accordingly to obtain the improved result in Appendix A.

**Theorem 4.3.** *Let $\mathcal{H}$ be a finite hypothesis class. A random ordering of $n$ examples is universally PAC-valid with error $\min\{\frac{n\varepsilon + \log \log n}{k}, 1\}$ and confidence $1 - \delta$, provided $n$ is large enough for $(\varepsilon, \delta)$-PAC learnability.*

This same argument can be extended to hypothesis classes with finite VC dimension. The proof is again deferred to the appendix due to space constraints.

**Theorem 4.4.** *Let $\mathcal{H}$ be a hypothesis class with $VC(\mathcal{H}) = d$. A random ordering of $n$ examples is universally PAC-valid with confidence $1 - \delta$ and error bounded by $O(\frac{n\varepsilon + \log \log n}{k})$ for any $d \leq k \leq n$.*

# 5 Main Result: Anytime-Valid Analysis

While the $O(\log \log n)$ result is strong, we demonstrate that the overhead factor is an artifact of the union-bound technique. Towards this, we now introduce a more direct analysis using the machinery of anytime-valid inference, which reveals that no such penalty is fundamentally necessary. This highlights the power of the confidence interval literature's machinery to resolve problems in our universal learning framework. These methods show that fixing a random permutation of the data is equivalent to randomly sampling $k$ points—a fact which will further inspire optimality questions in Section 6.

## 5.1 Constructing the Supermartingale and Applying Ville's Inequality

We can frame our analysis using a game-theoretic analogy. For any given hypothesis $h \in \mathcal{H}$, we can define a "null hypothesis":

$$H_0^{(h,k)} := err_{\mathcal{D}}(h) > \varepsilon_k,$$

or that $h$ is 'bad' for prefix size $k$. A skeptic can then place bets *against* this null hypothesis. This skeptic's bet is designed to form a non-negative supermartingale under $H_0^{(h,k)}$. If the bet amount grows significantly, it provides strong evidence to reject the null, meaning we can be confident that $h$ is 'good'.

To formalize this, we construct a test supermartingale $\left(M_k^{(h)}\right)_{k \geq 1}$ for each hypothesis $h \in \mathcal{H}$. A crucial observation is that a random permutation of a fixed dataset of size $n$ (where $n$ is sufficiently large) is equivalent to sampling without replacement from that finite population (conditioned on the sample complexity). This allows us to adapt powerful martingale constructions from the sampling without replacement literature (Waudby-Smith & Ramdas, 2020), such as the prior-posterior-ratio martingale. For the i.i.d. setting, a simple and powerful choice is a likelihood-ratio martingale.

We form a mixture martingale over the entire hypothesis class: $M_k = \sum_{h \in \mathcal{H}} \pi(h) M_k^{(h)}$, where $\pi(h)$ is a prior over $\mathcal{H}$ (e.g., uniform for a finite class). This mixture process $(M_k)_{k \geq 1}$ is also a non-negative supermartingale with $M_0 = 1$. We can now apply Ville's inequality directly to this single process. With probability at least $1 - \delta$, we have $M_k < 1/\delta$ for all $k \in \{1, \ldots, n\}$. If there existed some $k$ and a hypothesis $h_k$ that was consistent with the prefix $S_k$ but had high error, its corresponding martingale $M_k^{(h_k)}$ would be large. This would cause the mixture $M_k$ to become large, an event that Ville's inequality bounds with probability at most $\delta$. This line of reasoning leads to our main, improved theorems.

**Theorem 5.1** (Anytime-Valid Guarantee, Finite Class). *Let $\mathcal{H}$ be a finite hypothesis class. A random order of $n$ examples is universally PAC-valid with error $\epsilon_k \leq \frac{\log |\mathcal{H}| + \log(1/\delta)}{k}$ and confidence $1 - \delta$.*

*Proof.* We construct a single non-negative supermartingale and apply Ville's inequality to obtain a uniform bound over all prefix lengths $k$.

For each hypothesis $h \in \mathcal{H}$, consider testing the null hypothesis $H_0^{(h)} : err_{\mathcal{D}}(h) > \epsilon_k$ against the alternative $H_1^{(h)} : err_{\mathcal{D}}(h) = 0$, where we will set $\epsilon_k$ shortly. We can construct a likelihood-ratio process for a sequence of observations $(z_1, z_2, \ldots)$ as

$$M_t^{(h)} = \prod_{i=1}^{t} \frac{P(z_i | H_1^{(h)})}{P(z_i | err_{\mathcal{D}}(h) = \epsilon_k)}.$$

If $h$ is consistent with sample $z_i$, this ratio is $\frac{1}{1-\epsilon_k}$. If not, the numerator is 0. Thus, for a prefix $S_k$, $M_k^{(h)} = (1 - \epsilon_k)^{-k}$ if $h$ is consistent with $S_k$, and 0 otherwise. For any $h$ where $err_{\mathcal{D}}(h) > \epsilon_k$, this process is a non-negative supermartingale.

Now, define a mixture martingale over the entire class using a uniform prior $\pi(h) = 1/|\mathcal{H}|$:

$$M_k = \sum_{h \in \mathcal{H}} \pi(h) M_k^{(h)} = \frac{1}{|\mathcal{H}|} \sum_{h \text{ consistent with } S_k} (1 - \epsilon_k)^{-k}$$

This mixture process $(M_k)_{k \geq 1}$ is a non-negative supermartingale under the global null hypothesis that any hypothesis consistent with the data has error at least $\epsilon_k$. By Ville's inequality (Theorem 3.6), we have $P(\exists k : M_k \geq 1/\delta) \leq \delta$.

We now define the target error bound as $\epsilon_k = \frac{\log(|\mathcal{H}|/\delta)}{k}$. We prove by contradiction. Assume a "bad event" occurs: for some $k \in \{1, ..., n\}$, the learner returns a hypothesis $h_k$ that is consistent with $S_k$ but has true error $err_{\mathcal{D}}(h_k) > \epsilon_k$. If this event occurs, then at that prefix length $k$, the martingale $M_k$ must be at least:

$$M_k \geq \frac{1}{|\mathcal{H}|} M_k^{(h_k)} = \frac{1}{|\mathcal{H}|}(1 - \epsilon_k)^{-k}$$

We can lower-bound this term by taking the natural logarithm, using the inequality $-\log(1-x) \geq x$ for $x \in [0, 1)$, and exponentiating both sides to give $(1 - \epsilon_k)^{-k} \geq |\mathcal{H}|/\delta$. Substituting this back, we find that if the bad event occurs, then $M_k \geq \frac{1}{|\mathcal{H}|}\left(\frac{|\mathcal{H}|}{\delta}\right) = \frac{1}{\delta}$.

We have thus far shown that if a bad hypothesis is learned at any step $k$, it implies $M_k \geq 1/\delta$. But Ville's inequality tells us that the probability of the event $\{\exists k : M_k \geq 1/\delta\}$ is at most $\delta$. Therefore, the probability of a bad hypothesis being learned at any step is also at most $\delta$.

Thus, with probability at least $1 - \delta$, for all $k \in \{1, ..., n\}$, any hypothesis $h_k$ returned by a consistent learner on $S_k$ satisfies $err_{\mathcal{D}}(h_k) \leq \epsilon_k = \frac{\log(|\mathcal{H}|/\delta)}{k}$. Relating this to the standard sample complexity $n\epsilon = O(\log|\mathcal{H}| + \log(1/\delta))$, we see the bound is $O(\frac{n\epsilon}{k})$. $\qquad\square$

As before, we extend this logic to the case of infinite hypothesis class with finite $\text{VC}(\mathcal{H})$. The full proof details are deferred to the appendix due to space constraints.

**Theorem 5.2** (Anytime-Valid Guarantee, Infinite VC-dim Class). *Let $\mathcal{H}$ be a hypothesis class with $VC(\mathcal{H}) = d$. A random ordering of $n$ examples is universally PAC-valid with error $\epsilon_k = O(\frac{d + \log(1/\delta)}{k}) = O(\frac{n\epsilon}{k})$ for any $d \leq k \leq n$ with confidence $1 - \delta$.*

The proofs for these theorems rely on the properties of the constructed supermartingale and a single application of Ville's inequality. The analysis correctly models the dependencies between prefixes from first principles, demonstrating that the logarithmic overhead factors from the union-bound analysis are not fundamental to the problem but are artifacts of that specific proof technique.

## 6 DISCUSSION

In this work, we formally defined the universal data ordering problem within the PAC framework and provided a rigorous baseline analysis for the performance of a random permutation, a provably optimal task-agnostic method. Our goal was to establish whether this simple, natural approach could serve as a strong foundation for this problem, and our results show that it is surprisingly effective.

Our investigation yielded progressively stronger results, highlighting the power of modern sequential analysis tools. The warm-up analysis, based on classical union bounds, established that random orderings are surprisingly robust, suffering only a minor $O(\log \log n)$ penalty. However, our main contribution is the tighter analysis via test supermartingales and Ville's inequality. This approach not only removes the logarithmic overhead, achieving the optimal statistical rate, but also provides a more profound understanding of the problem's structure.

The primary limitation of our work is that the analysis centers on random orderings. While we show such orderings are surprisingly effective, they are unlikely to retain optimality when restricted to subclasses of tasks (rather than fully agnostic). Our work opens several avenues for future research:

- **Designing Optimal Orderings:** The martingale framework suggests a new direction for designing structured orderings. Could one design an ordering algorithm that actively tries to maximize the growth of martingales corresponding to incorrect hypotheses, thereby falsifying them more quickly? This connects to ideas in "safe testing" and could lead to provably better-than-random orderings (Ramdas et al., 2023).
- **Beyond PAC Learning:** The anytime-valid perspective on data ordering could be extended to other sequential learning settings, such as online convex optimization or regret minimization in bandits, where the sequence of data presentation is crucial.

- **PAC-Bayes Confidence Sequences:** A more advanced direction is to move beyond high-probability bounds to full distributional guarantees. One could leverage the rich literature on PAC-Bayes analysis to construct anytime-valid PAC-Bayes bounds, or confidence sequences, for the risk of the hypothesis at each prefix $k$ (Chugg et al., 2023; Rodriguez-Galvez et al., 2024). This would provide a posterior distribution over the possible error values, offering a more complete characterization of uncertainty.

Overall, our analysis establishes rigorous baseline guarantees for universal orderings, highlights the surprising effectiveness of random permutations, and connects this fundamental problem in learning theory to the powerful and general framework of anytime-valid inference. Future work should explore methods for constructing orderings that outperform random permutations.

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

# A  OMITTED PROOFS

## A.1  UNION BOUND PROOFS

*Proof.* To show universal PAC-validity, we must demonstrate that with probability at least $1-\delta$, for every prefix $S_k$ of size $k \in \{1, \ldots, n\}$, the hypothesis $h_k = \mathcal{A}(S_k)$ has error at most some $\varepsilon_k$.

Let $B_k$ be the "bad event" for a prefix of size $k$:

$$B_k := \{\exists h \in \mathcal{H} \text{ consistent with } S_k \text{ but } err_{\mathcal{D}}(h) > \varepsilon_k\}$$

We bound the probability of the union of these bad events over all $k$.

$$P(\cup_{k=1}^{n} B_k) \leq \sum_{k=1}^{n} P(B_k)$$

For a fixed hypothesis $h \in \mathcal{H}$ with $err_{\mathcal{D}}(h) > \varepsilon_k$, the probability that it is consistent with $k$ i.i.d. samples is $(1 - err_{\mathcal{D}}(h))^k < (1 - \varepsilon_k)^k \leq e^{-k\varepsilon_k}$. By a union bound over all hypotheses in $\mathcal{H}$, we get $P(B_k) \leq |\mathcal{H}|e^{-k\varepsilon_k}$.

To ensure the total failure probability is at most $\delta$, we allocate a failure probability of $\delta/n$ to each prefix size $k$. Thus, we require $P(B_k) \leq \delta/n$ for each $k$.

$$|\mathcal{H}|e^{-k\varepsilon_k} \leq \frac{\delta}{n}$$

Solving for $\varepsilon_k$:

$$-k\varepsilon_k \leq \log\left(\frac{\delta}{n|\mathcal{H}|}\right) = \log\left(\frac{\delta}{|\mathcal{H}|}\right) - \log n$$

$$\varepsilon_k \geq \frac{\log(|\mathcal{H}|/\delta) + \log n}{k}$$

This proves the theorem. $\qquad\qquad\qquad\qquad\qquad\qquad\qquad\qquad\qquad\qquad\qquad\qquad\qquad$ $\square$

In resolving this additional $\log n$ factor to $\log \log n$ as discussed in Section 4.2, we first prove the critical "stability" lemma.

**Lemma A.1.** *Let $h_k$ be a hypothesis consistent with the first $k$ samples of a random ordering, $S_n$, drawn i.i.d. from a distribution $\mathcal{D}$, and assume $err_{\mathcal{D}}(h_k) \leq \varepsilon_k$. Then for any $\eta \in (0, 1]$, the probability that $h_k$ is also consistent with the next $\eta k$ examples in the sequence is at least $(1-\varepsilon_k)^{\eta k}$.*

*Proof of Lemma 4.2.* The samples $z_{k+1}, \ldots, z_{k+\eta k}$ are i.i.d. draws from $\mathcal{D}$. The probability that $h_k$ is consistent with a single new sample $z_i$ is $1 - err_{\mathcal{D}}(h_k)$, which is at least $1 - \varepsilon_k$. Since the samples are independent, the probability of being consistent with all $\eta k$ new samples is $(1 - err_{\mathcal{D}}(h_k))^{\eta k} \geq (1 - \varepsilon_k)^{\eta k}$. $\qquad\qquad\qquad\qquad$ $\square$

Armed with this lemma, we proceed to revise the Proof of Theorem 4.1 with a more careful accounting of accumulated errors to obtain the main result of the warm-up section.

*Proof of Theorem 4.3.* Instead of a union bound over all $n$ prefixes, we use a more efficient union bound over a set of geometrically spaced "anchor points." Let $\eta \in (0, 1]$ and define the anchor points as $k_j = \lfloor (1+\eta)^j \rfloor$ for $j = 0, 1, \ldots, L$, where $L = \lceil \log_{1+\eta} n \rceil$. The number of anchor points is $L + 1 = O(\log n)$.

Let $B_j$ be the bad event at anchor point $k_j$:

$$B_j := \{\exists h \in \mathcal{H} \text{ consistent with } S_{k_j} \text{ but } err_{\mathcal{D}}(h) > \varepsilon_{k_j}\}.$$

We apply a union bound over these $L + 1$ events, allocating a failure probability of $\delta/(L+1)$ to each.

$$P(B_j) \leq |\mathcal{H}|e^{-k_j \varepsilon_{k_j}} \leq \frac{\delta}{L+1}$$

Solving for $\varepsilon_{k_j}$:

$$\varepsilon_{k_j} \geq \frac{\log(|\mathcal{H}|(L+1)/\delta)}{k_j} = \frac{\log(|\mathcal{H}|/\delta) + \log(L+1)}{k_j}$$

For any $k \in [d, n]$, let $j$ be such that $k_j \leq k < k_{j+1}$. The hypothesis $h_k$ is trained on $S_k$. With probability at least $1 - \delta$, for all $j = 0, \ldots, L$, any hypothesis $h'_{k_j}$ consistent on $S_{k_j}$ has $err_{\mathcal{D}}(h'_{k_j}) \leq \varepsilon_{k_j}$. By Lemma A.1, $h_{k_j}$ is also consistent on $S_k$ with high probability. In the realizable setting, this implies that the hypothesis $h_k$ found by a consistent learner on $S_k$ must also satisfy $err_{\mathcal{D}}(h_k) \leq \varepsilon_{k_j}$. Since $k \geq k_j$, the bound thus holds for all intermediate $k$. $\qquad\square$

Lastly, we prove a matching result in the finite VC dimension setting.

*Proof of Theorem 4.4.* The proof structure is identical to that of Theorem 4.3 for finite classes, but we adapt it for a hypothesis class $\mathcal{H}$ with a finite VC-dimension $d = VC(\mathcal{H})$. The key difference is that the number of ways the hypothesis class can label a sample of size $k$ is no longer bounded by $|\mathcal{H}|$, but by the growth function, $\tau_{\mathcal{H}}(k)$. For $k \geq d$, Sauer's Lemma provides the bound $\tau_{\mathcal{H}}(k) \leq \left(\frac{ek}{d}\right)^d$.

As in the proof of Theorem 4.3, we define a set of $L + 1 = O(\log n)$ geometrically spaced anchor points $k_j = \lfloor (1 + \eta)^j \rfloor$. Our goal is to ensure that with high probability, for every anchor point $k_j$, any hypothesis consistent with the prefix $S_{k_j}$ has a low true error.

Let $B_j$ be the bad event at anchor point $k_j$:

$$B_j := \{\exists h \in \mathcal{H} \text{ consistent with } S_{k_j} \text{ but } err_{\mathcal{D}}(h) > \epsilon_{k_j}\}$$

By applying a union bound over the $\tau_{\mathcal{H}}(k_j)$ possible labelings of the sample $S_{k_j}$, the probability of this bad event is bounded by:

$$P(B_j) \leq \tau_{\mathcal{H}}(k_j) e^{-k_j \epsilon_{k_j}}$$

We apply a union bound over all $L + 1$ anchor points, allocating a failure probability of $\delta/(L+1)$ to each. To ensure the total probability of failure is at most $\delta$, we require for each $j$:

$$\tau_{\mathcal{H}}(k_j) e^{-k_j \epsilon_{k_j}} \leq \frac{\delta}{L+1}$$

Solving for the error $\epsilon_{k_j}$:

$$-k_j \epsilon_{k_j} \leq \log\left(\frac{\delta}{(L+1)\tau_{\mathcal{H}}(k_j)}\right)$$

$$\epsilon_{k_j} \geq \frac{\log(\tau_{\mathcal{H}}(k_j)) + \log((L+1)/\delta)}{k_j}$$

Now, we substitute the bound for the growth function and the fact that $L + 1 = O(\log n)$:

$$\epsilon_{k_j} \geq \frac{d\log(ek_j/d) + \log(O(\log n)) + \log(1/\delta)}{k_j} = O\left(\frac{d\log(k_j) + \log\log n + \log(1/\delta)}{k_j}\right)$$

This bound holds simultaneously for all anchor points $j = 0, ..., L$ with probability at least $1 - \delta$.

For any intermediate prefix length $k$ such that $k_j \leq k < k_{j+1}$, the prefix $S_{k_j}$ is a subset of $S_k$. If our guarantee holds at anchor point $k_j$, it means no hypothesis $h$ with error $err_{\mathcal{D}}(h) > \epsilon_{k_j}$ is consistent with $S_{k_j}$. Therefore, no such hypothesis can be consistent with the larger sample $S_k$. This implies that the hypothesis $h_k$ returned by a consistent learner on $S_k$ must have an error $err_{\mathcal{D}}(h_k) \leq \epsilon_{k_j}$. Since $k \geq k_j$, its error is bounded by:

$$err_{\mathcal{D}}(h_k) \leq \epsilon_{k_j} = O\left(\frac{d\log(k) + \log\log n}{k_j}\right)$$

Using the standard sample complexity definition where $n\epsilon = O(d + \log(1/\delta))$, this bound simplifies to $O(\frac{n\epsilon + \log\log n}{k})$. This completes the proof. $\qquad\square$

## A.2 ANYTIME-VALID PROOFS

*Proof of Theorem 5.2.* The proof is analogous to that of Theorem 5.1, but we must handle the fact that the hypothesis class $\mathcal{H}$ is infinite. We achieve this by replacing the uniform prior over $\mathcal{H}$ with an adaptive prior that considers only the set of behaviors of $\mathcal{H}$ on the observed data prefix $S_k$. The size of this set of behaviors (dichotomies) is bounded by the growth function $\tau_{\mathcal{H}}(k)$.

Let $\epsilon_k = \frac{\log(\tau_{\mathcal{H}}(k)) + \log(1/\delta)}{k}$. For each $k$, we construct a mixture martingale to test the global null hypothesis that any hypothesis consistent with $S_k$ has an error of at least $\epsilon_k$.

Let $\Pi_{\mathcal{H}}(S_k)$ be the set of all possible labelings (dichotomies) that the class $\mathcal{H}$ can induce on the sample $S_k$. We know that $|\Pi_{\mathcal{H}}(S_k)| \leq \tau_{\mathcal{H}}(k)$. We can define a mixture martingale with a uniform prior over these dichotomies:

$$M_k = \frac{1}{|\Pi_{\mathcal{H}}(S_k)|} \sum_{h \in \Pi_{\mathcal{H}}(S_k)} M_k^{(h)}$$

where $M_k^{(h)} = (1 - \epsilon_k)^{-k}$ is the test martingale for a single hypothesis (labeling) $h$ against the null $err_{\mathcal{D}}(h) = \epsilon_k$. As the size of the set of dichotomies can change with $k$, this is an adaptive mixture. This process $(M_k)_{k \geq d}$ remains a non-negative supermartingale. We can now apply Ville's inequality, which states that $P(\exists k \geq d : M_k \geq 1/\delta) \leq \delta$.

We proceed by contradiction. Assume a "bad event" occurs: for some $k \geq d$, the learner returns a hypothesis $h_k$ that is consistent with $S_k$ but has true error $err_{\mathcal{D}}(h_k) > \epsilon_k$.

If this bad event occurs at step $k$, then the martingale $M_k$ is lower-bounded by the term corresponding to the observed labeling induced by $h_k$:

$$M_k = \frac{1}{|\Pi_{\mathcal{H}}(S_k)|} \sum_{h \in \Pi_{\mathcal{H}}(S_k), \text{consistent}} (1 - \epsilon_k)^{-k} \geq \frac{1}{\tau_{\mathcal{H}}(k)} (1 - \epsilon_k)^{-k}$$

We want to show that this event implies $M_k \geq 1/\delta$. This requires showing that $\frac{1}{\tau_{\mathcal{H}}(k)} (1 - \epsilon_k)^{-k} \geq 1/\delta$. Taking the logarithm of the desired inequality $(1 - \epsilon_k)^{-k} \geq \tau_{\mathcal{H}}(k)/\delta$:

$$-k \log(1 - \epsilon_k) \geq \log(\tau_{\mathcal{H}}(k)/\delta)$$

Using the fact that $-\log(1 - x) \geq x$ for $x \in [0, 1)$, it is sufficient to show:

$$k\epsilon_k \geq \log(\tau_{\mathcal{H}}(k)) + \log(1/\delta)$$

By our definition of $\epsilon_k = \frac{\log(\tau_{\mathcal{H}}(k)) + \log(1/\delta)}{k}$, this condition is met exactly.

Therefore, the occurrence of a "bad event" at step $k$ implies that $M_k \geq 1/\delta$. Since the probability of the latter is bounded by $\delta$ for all $k$ simultaneously, the probability of a bad event ever occurring is also at most $\delta$.

This means, with probability at least $1 - \delta$, for all $k \in \{d, ..., n\}$, any hypothesis $h_k$ returned by a consistent learner on $S_k$ satisfies:

$$err_{\mathcal{D}}(h_k) \leq \epsilon_k = \frac{\log(\tau_{\mathcal{H}}(k)) + \log(1/\delta)}{k}$$

Substituting the bound $\tau_{\mathcal{H}}(k) \leq (ek/d)^d$, we get:

$$err_{\mathcal{D}}(h_k) \leq \frac{d \log(ek/d) + \log(1/\delta)}{k} = O\left(\frac{d \log k + \log(1/\delta)}{k}\right)$$

This is the standard, fixed-sample-size PAC bound for a VC class. Our anytime-valid analysis proves that it holds uniformly for all prefixes $k \geq d$. Relating this to the sample complexity $n\epsilon = O(d + \log(1/\delta))$, the bound is $O(\frac{n\epsilon}{k})$. $\qquad\square$

## A.3 OPTIMALITY OF RANDOM ORDERING

In this section, we formalize the comparison between deterministic and random orderings in the task-agnostic setting.

### A.3.1 Deterministic Orderings Suboptimality

We show that for any deterministic ordering algorithm, there exists a realizable PAC learning task where the algorithm fails to provide universal guarantees.

**Theorem A.2.** *Let $\mathcal{A}_{det}$ be a deterministic ordering algorithm. There exists a domain $\mathcal{X}$, a distribution $\mathcal{D}$, and a hypothesis class $\mathcal{H}$ such that a consistent learner returns a hypothesis with high error on the prefix samples from $\mathcal{A}_{det}$.*

*Proof.* Let $\mathcal{X} = \{x_A, x_B\}$ and let the true distribution $\mathcal{D}$ be defined such that $\mathbf{Pr}(x_A) = \epsilon$ and $\mathbf{Pr}(x_B) = 1 - \epsilon$ for a small $\epsilon > 0$. Let the target concept $c$ be $c(x_A) = 0$ and $c(x_B) = 1$.

We define the hypothesis class $\mathcal{H} = \{h_0, h^*\}$ where $h^*(x) = c(x)$ and $h_0(x) = 0$ for all $x \in \mathcal{X}$. Note that the problem is realizable because $h^* \in \mathcal{H}$ that has VC dimension 1?.

Assume without loss of generality that $\mathcal{A}_{det}$ orders $x_A$ before $x_B$. We further assume to have a consistent learner who draws an $n$ sample set from $\mathcal{D}$. Let $E$ be the event that this sample set contains at least one $x_A$ sample: $\mathbf{Pr}[E] = 1 - (1 - \epsilon)^n$ which tends to 1 as $n \to \infty$. Conditional on $E$, we have that the sampled set contains $k \geq 1$ instances of $x_A$ and $n - k$ of $x_B$. The algorithm applied to this set produces an ordering where the first $k$ elements are $x_A$. For the prefixes up to $k+1$, the learner only sees samples with label 0. Thus, both $h^*$ and $h_0$ are consistent with this prefix. A consistent learner may therefore return $h_0$. However, the true error of $h_0$ on the distribution is:

$$err_{\mathcal{D}}(h_0) = \mathbf{Pr}(x_B) \cdot \mathbb{I}[h_0(x_B) \neq 1] = 1 - \epsilon$$

Thus, for the prefix where only $x_A$ is observed, the learner suffers maximal error, proving that deterministic orderings are not universally PAC-valid. $\square$

### A.4 Minimax Optimality of Uniform Random Permutations

We explicitly show that the uniform random ordering is the unique minimax optimal strategy for task-agnostic ordering.

**Theorem A.3.** *The Uniform Random Ordering minimizes the maximum risk of failing to include a critical sample in a prefix of size $k$.*

*Proof.* Let $S_k$ be the set of indices in the prefix of size $k$ for a drawn dataset $S \sim \mathcal{D}$. Consider a similar task to the construction in Theorem A.2 where the concept is revealed by a single ideal index $i^* \in \{1, \ldots, n\}$. For any randomized ordering algorithm defined by a distribution $\mathcal{P}$ over permutations, the expected size of the prefix is exactly $k$. By linearity of expectation:

$$\sum_{i=1}^{n} \mathbf{Pr}_{\pi \sim \mathcal{P}}[i \in S_k] = \mathbb{E}[|S_k|] = k$$

This implies that the arithmetic mean of the inclusion probabilities is exactly $k/n$ regardless of the algorithm. An adversary, observing the algorithm $\mathcal{P}$, will select the index $i^*$ to minimize its probability of inclusion:

$$\min_i \mathbf{Pr}[i \in S_k]$$

Since the minimum of a set of numbers is bounded above by their average, we have:

$$\min_i \mathbf{Pr}[i \in S_k] \leq \frac{1}{n} \sum_{i=1}^{n} \mathbf{Pr}[i \in S_k] = \frac{k}{n}$$

This upper bound is achieved if and only if all inclusion probabilities are equal. Thus, any non-uniform distribution implies there exists some index $j$ such that $\mathbf{Pr}[j \in S_k] < k/n$, which the adversary will exploit to maximize the failure rate. The uniform random ordering is therefore the unique strategy that assigns $\mathbf{Pr}[i \in S_k] = k/n$ for all $i$, and thus it is minimax optimal. $\square$

