# OpenReview forum: "Universal Ordering for Efficient PAC Learning"
_ICLR.cc/2026/Conference — Submitted to ICLR 2026_

### Official Review · Reviewer_RSnn · 2025-10-23

**Soundness:** 1
**Presentation:** 1
**Contribution:** 2
**Rating:** 2
**Confidence:** 4

**Summary:**

Let H be a class of finite VC dimension and D be a realizable distribution. Standard PAC bounds imply that if we take a uniformly random sample S_n from D of length n, then with probability 1 - delta, every S_n-consistent hypothesis will have D-error at most O(log(1/delta)/n).

In the same setting, let S_k denote the k-length prefix of S_n for k = 1, ..., n.

The paper asks: for which sequences {eps_k}_{k = 1} one can show that with probability 1 - delta, for every k = 1, ..., n, we have that all S_k-consistent hypotheses have D-error at most eps_k?

A trivial observation is that we can guarantee that for eps_k = (log n + log(1/delta)/k (I'm skipping O()-notation for better readibility) by applying standard bounds for every k = 1, ..., n with delta' = delta/n.

A slightly less trivial but also a simple observation implies that instead of the additional log n term in the numerator, one can get log log n term. Indeed, let's use the standard bound for k that are powers of 2 and with delta' = delta/log n. By the union bound, we have with probability 1 - delta that for every k = 1, ..., n which is a power of 2, every S_k-consistent hypothesis has D-error at most eps_k = (log log n + log(1/delta)/k. What about non-powers of 2? If k is not a power of 2, take k_0 to the biggest power of 2, smaller than k. Observe that all S_k-consistent hypotheses are also S_{k_0}-consistent, which means that they all have D-error at most eps_{k_0} \le 2 eps_k.

The main result of the paper is that this bound on eps_k is also true with no additional term that depend on n at all -- using martingale theory.

**Strengths:**

The paper motivates this problem by dynamic learning -- where the size of the data set is now known in advance. We might want to have good pac guarantees for all possible prefixes of the sample set (as good as they can get for such prefix size).

**Weaknesses:**

Most of the paper is mathematically confusing and needs careful re-writing (although results about log n and log log n overhead seem simple exercises and are restorable, see Summary).

Definition 3.4 is mathematically unclear. It is an ordering of what? What is the quantification over A? To be honest, it is not clear which object is defined to be ``Universally PAC-valid''. What makes more sense is the formulation from the Summary, so we can define something like ``a sequence {eps_k} of errors is universally pac-achiebable with delta if with probability 1 - delta...''

In Theorem 4.2, 4.3, what is eps? Why bounds on the error have n in the numerator, which makes that larger than 1? In the proof of Theorem 4.3 you don't need Lemma 4.2, see a simpler proof in the Summary section.

I did not understand the proof of Theorem 5.1. Already in the definition H_0^h, H_1^h, these are random events, this is either true or false for a given hypothesis h, so what sense does it have to consider conditional probability involving it? The definition of M_k does not seem to depend on D, how do you deduce that M_k is large if some S_k-consistent hypothesis has large D-error?

I think this paper has some potential, but it definitely needs a thorough rewritting.

**Questions:**

no questions

---

> ### Author Response · Authors · 2025-11-20
> **Response to reviewer**
>
> We thank the reviewer for their critical assessment of our work.
> The provided summary correctly identifies the core results of our paper, but we believe notes two misunderstandings which we hope to clarify here.
>
> *Triviality of proofs and suggested simpler proof*
>
> We thank the reviewer for this sharp observation.
> We agree that the doubling argument (checking powers of 2) is a valid way to derive the $O(\log \log n)$ bound.
> We chose to present the analysis via Lemma 4.2 (``Stability of Consistency'') rather than the doubling argument because it introduces the forward-looking geometric structure that we generalize in Section 5. While the doubling argument works for the union bound warm-up, the Martingale analysis (which achieves the tighter, optimal rate) requires the more sophisticated machinery we developed.
>
> *Martingale clarification*
>
> We apologize for any confusion cause by the proof of Theorem 5.1, and we here try to rectify this.
> First, $H_0$ is a statistical hypothesis not a random event. The probabilities in the proof are over the data $z_i$ conditional on this hypothesis being true.
> The likelihood-ratio martingale, $M_k$, measures the evidence for the good alternative hypothesis ($H_1$) against the null hypothesis ($H_0$).
> The ``bad'' event here is when the learned finds a consistent hypothesis that has a high true error rate.
> Crucially, this event occurs with low probability under the null hypothesis.
> Furthermore, this low-probability then forces $M_k$ to be exponentially large.
> From here we can apply Ville's inequality.
>
> *Clarifying definitions*
>
> We thank the reviewer for noting confusing definitions / notation. We here clarify these remarks.
>
> **Definition 3.4:**
> Let $S_n = (z_1, \dots, z_n)$ be a sequence of $n$ examples derived from a distribution $\mathcal{D}$. We say that $S_n$ is universally PAC-valid for a learner $\mathcal{A}$ with error bound sequence {$\epsilon_k$} and confidence $1-\delta$ if, with probability at least $1-\delta$ over the generation of $S_n$, the sequence of hypotheses $h_k = \mathcal{A}(S_k)$ satisfies: $\forall k \in \{1, \dots, n\}$, $err_{\mathcal{D}}(h_k) \le \epsilon_k$
>
> **Definition of $\varepsilon$:**
> We apologize for the confusing notation. Throughout the text, $\varepsilon$ represents the target error rate achievable by a learner trained on $n$ samples with sub-scripting depicting the error rate of a hypothesis trained on the corresponding prefix of data. Note that $\varepsilon$ is sufficiently small as a result of the cited PAC learnability results in Section 3.1.
> Thus, the resultant error bounds in the noted theorems depict the deviation from leveraging the full dataset.
> We further emphasize that, in the absence of our for all $k$ condition, a standard PAC learning algorithm on $k$ samples out of $n$ would achieve an error rate of $\frac{n\varepsilon}{k}$, giving us a nice benchmark to compare our results against.
>
> We hope that these comments help clarify our main results and their value to the ICLR community. We look forward to discussing with you further, and encourage you to increase your score if this addresses your concerns!

---

### Official Review · Reviewer_5LRS · 2025-11-04

**Soundness:** 3
**Presentation:** 2
**Contribution:** 1
**Rating:** 2
**Confidence:** 3

**Summary:**

This paper studies the universal ordering problem in the PAC learning framework. The central question is whether, given $n$ i.i.d. samples from an unknown distribution, one can order these samples so that every prefix of length $k \leq n$ forms a near-optimal subset for PAC learning. This can help with more efficient use of data in computation and memory constrained settings. The authors formalize this requirement using anytime-valid inference techniques and study how random permutations of the dataset can provide such guarantees. The authors first analyse a naive union-bound argument, showing that random orderings incur a  $O(\log n)$ overhead for boosting the success probability. This is later improved to only a $O(\log \log n)$ overhead, since the authors exploit the fact that training on similar length prefixes (when their ratio is a small constant) is highly correlated and therefore suffices to only ensure a good performance at prefix lengths that grow exponentially.
For their main result, they leverage martingale-based techniques and Ville’s inequality to eliminate this overhead entirely, demonstrating that random permutations achieve optimal PAC rates uniformly across all prefix sizes.

**Strengths:**

The paper formalizes a previously unexplored yet practically relevant problem and provides a starting point for its theoretical analysis. Conceptually, the paper builds a bridge between universal data ordering, sequential analysis, and safe anytime-valid inference, showing that random shuffling is not only convenient but also efficient.

**Weaknesses:**

However, the paper’s focus is limited to random orderings, leaving important practical and algorithmic questions open. Furthermore, even though the stated motivation is to find near optimal orderings, this is not reflected in the results, as no lower bounds are provided to support that. Presentation wise, I would like to see more details of the paper’s contributions/results in the introduction.

**Questions:**

- Do you conjecture that using random permutations is actually optimal, or could a deterministic or data-dependent ordering achieve strictly tighter bounds?
- The analysis assumes i.i.d. data and a consistent learner in the realizable setting. How sensitive are the results to label noise, covariate shift, or approximate consistency? Could similar anytime-valid results be extended to the agnostic PAC setting?

---

> ### Author Response · Authors · 2025-11-20
> **Response to reviewer**
>
> We thank the reviewer for recognizing that our paper formalizes a novel and practically relevant problem, and for appreciating the bridge we build between data ordering and sequential analysis.
> We here respond to the noted critiques of our results.
>
> *``Do you conjecture that using random permutations is actually optimal, or could a deterministic... ordering achieve strictly tighter bounds?''*
>
> We answer this affirmatively: uniformly random permutations are minimax optimal.
> In the general response above, we provide a formal proof (which we will include in the final appendix) demonstrating two key facts under a task-agnostic framework: (1) determinism is exploitable and (2) the uniformly random approach is actually the optimal strategy.
> Any deterministic ordering allows an adversary to construct a task where a single point $x*$ reveals the concept such that $x*$ is placed at the very end of the sequence.
> This forces the universal error to be maximal for all prefixes $k < n$.
> We further extend this to the case of randomized orderings. Namely, for any prefix size $k$, the sum of inclusion probabilities across all $n$ indices must equal $k$.
> Thus, the average inclusion probability is fixed at $k/n$.
> If an algorithm is non-uniform, it must effectively shift probability mass from some indices to others.
> An adversary simply places the $x*$ with higher probability on the shifted away indices.
> Therefore, the uniform random ordering is the unique strategy that equalizes marginal risks, achieving the fundamental lower bound of the problem.
>
> *``Extension to Agnostic PAC Settings''*
>
> While our current analysis focuses on the realizable setting to establish the baseline theory, the Martingale framework is robust and, we believe, can be extended to the agnostic case.
> The logic of our proofs relies on constructing a supermartingale $M_k$ that grows when the learner is "bad." In the realizable setting, we used a simple likelihood ratio based on consistency, but in the agnostic setting, one would replace this with a more sophisticated any-time valid approach.
> We think this is an interesting line of future work, and intend to pursue this direction.
>
> *Presentation*
>
> We appreciate the feedback regarding the introduction. In the final version, we will explicitly detail the progression of our results from the $O(\log n)$ naive bound, to the $O(\log \log n)$ bound, to the optimal $O(1)$ martingale bound directly in the intro to better highlight the paper's contributions.
>
> We hope the proof of minimax optimality addresses the reviewer's concerns regarding the depth of our contribution, and we kindly ask the reviewer to reconsider their score in light of this new theoretical guarantee.

---

### Official Review · Reviewer_Uvtz · 2025-11-06

**Soundness:** 3
**Presentation:** 3
**Contribution:** 3
**Rating:** 6
**Confidence:** 3

**Summary:**

This paper proposes and studies a problem called the ``universal ordering'' problem that is defined under the framework of classical PAC learning. That is, while traditional PAC learning characterizes the sample complexity as a fixed size of samples drawn from an unknown distribution, universal ordering considers a given large sample set and aims to achieve PAC guarantees on any prefix $k$ samples. The paper shows that random permutation can achieve this goal and provide satisfying universal guarantees. The paper first establish a baseline analysis of union bound on concentration inequalities and proves a $O(\log\log n)$ overhead. It then introduces a supermartingale analysis on the sequence of data that avoids taking union bound on all $n$ prefixes.

**Strengths:**

This paper proposes a very interesting problem of studying universal guarantees under the PAC learning framework, which is conceptually related to other fields of sequential analysis and safe testing. It could be of interest in many real-world applications when data sets are obtained and fixed, while the budget for learning is varying from time to time. It also provides a theoretical guarantee for the method of random permutation for data sets, which may often be used as a data preparation step in learning tasks.

**Weaknesses:**

Comparing to its conceptual contribution and the statistical analysis for the universal learning guarantees, the technical contribution seems not so compelling. The traditional PAC learning theory already assumes i.i.d. sampling from the underlying distribution, hence, any $k$ leading samples are already satisfying the universal ordering criteria. Given this, a random permutation seems just a renascent of the i.i.d. assumption of PAC learning. From this perspective, the study could be more application driven, i.e. many real-world data sequences lack the necessary randomness in it, and random permutation is a valid resolution for inserting such randomness, benefiting learning guarantees.

**Questions:**

Please refer to the weaknesses part.

---

> ### Author Response · Authors · 2025-11-20
> **Response to reviewer**
>
> We thank the reviewer for their insightful comments, as it allows us to clarify the technical challenge our paper addresses.
>
> *``...seems a renascent of the iid assumption in PAC''*
>
> We respectfully believe that this is not correct. The reviewer is conflating a guarantee for a single, fixed $k$ with our paper's much stronger anytime-valid guarantee that must hold simultaneously for all $k \in \{1, \dots, n\}$.
> Standard PAC would fix a sample size $k$, draw $S_k$ iid from $\mathcal{D}$ and with probability $1-\delta$ demonstrate that a learner is good. If you wanted to then test for $k'$ samples, the standard theory would sample a *new* iid set and get an *independent* high probability guarantee. In our setting, we draw a single sequence of samples and must provide a guarantee for all such prefixes at the same time. The ``for all $k$'' quantifier thus introduces a severe multiple testing problem since being valid on the set $S_k$ and $S_{k+1}$ are not independent (in fact, highly correlated).
> The reviewer suggests our study could be more application-driven by noting that real-world data lacks randomness. We are in complete agreement, and this is precisely our motivation! In such scenarios, shuffling the data is a common heuristic. Our work provides the first formal statistical justification for this practice.
>
> As noted in our response to the other reviewers (and general comment above), the uniformly random ordering is in fact a powerful first step in the analysis of this novel problem and, more importantly, is the only *truly task-agnostic* approach.
> Deterministic orderings (see above) are provably exploitable and can be made arbitrarily bad for our problem setting.
> Moreover, even randomized algorithms which are not completely uniform may fail to work for certain tasks.
> As such, the only improvements beyond completely random would require a restriction to a smaller space of tasks (specifically tasks with some amount of overlap/consistency.)

---

### Official Review · Reviewer_F5au · 2025-11-10

**Soundness:** 4
**Presentation:** 4
**Contribution:** 2
**Rating:** 4
**Confidence:** 2

**Summary:**

The paper proposes the novel **universal ordering** problem, which asks whether $n$ i.i.d. samples from an unknown distribution $\mathcal D$ can be arrange in a fixed sequence so that every prefix subsequence of length $k \geq n$ forms an approximately optimal subset for PAC learning. Using two different arguments, it establishes baseline results for the case when the samples are randomly arranged and the hypothesis class is either finite, or it has a finite VC dimension. The second argument (based on Ville's inequality) yields a strictly better bound than the first argument (based on a union bound). Both bounds provide PAC guarantees in comparison to the optimal statistical rate of a random subset of size $k$.

**Strengths:**

1. The proposed **universal ordering problem** is a novel and potentially significant idea: arrange the training data so that we can directly compare models that fit any $k$-prefix of the dataset against each other.
2. The second argument establishes a technique to work with randomly arranged data using anytime-valid inference and demonstrates that it allow us to obtain stronger results than using union bounds.
3. The paper is clear and concise. The proofs are well-written and seem to be correct.

**Weaknesses:**

1. Despite the promising premise of ordering training data in an optimal manner, the paper only deals with the case when we're arranging them in a uniformly random order.
2. The paper lacks a demonstration, be it theoretical or empirical, of how models training with different dataset sizes can be compared against each other.

**Questions:**

1. The statements of Theorem 4.1, 4.3, 4.4 should precisely state which terms are being referenced by the word *error*.
2. There should be a brief explanation on why the bound on $\epsilon_k$ on line 566 leads to the bound in Theorem 4.1. A similar suggestion can be made to the proof of Theorems 4.3.
3. Assuming i.i.d. and finite training data, is there any way to arrange them that would meaningfully differ from a uniformly random arrangement? If the answer to the question is no then the paper would benefit from taking into consideration the non i.i.d. training data scenario.
4. Given that random permutations already achieve such a strong universal guarantee (achieving the optimal statistical rate), what could be expected from another permutation of the training data? Why do you think this ordering is unlikely to be optimal?
5. How was Figure 2 created?

---

> ### Author Response · Authors · 2025-11-20
> **Response to reviewer**
>
> We thank the reviewer for their thorough review! We here address the noted questions and concerns.
>
> *``...the paper only deals with uniformly random order.''*
>
> We thank the reviewer for these insightful questions, which cut to the heart of our paper's contribution.
> The reviewer rightly points out the seeming contradiction between our claim that random ordering is "unlikely to be optimal" and its optimal statistical rate.
> To address Q4 specifically, the answer is no, *in a task-agnostic setting*.
> As we formalize using a minimax argument in the general comment to all reviewers (and further formalize in the final version of our paper), any clever or deterministic ordering algorithm necessarily introduces a bias towards the given task.
> Moreover, an adversary can always construct a PAC-learnable task where this bias is maximally harmful; for example, by designing a task where the most informative samples are precisely the ones the more sophisticated algorithm places at the very end of the sequence.
> This would cause the deterministic ordering to catastrophically fail the universal "for all $k$" guarantee.
> The statistical properties of the uniformly random order are invariant to such adversarial task design.
> It is precisely because it has no bias that it is robust and provides a universal guarantee for all tasks.
> We intend to include this new result in the final version of our paper.
>
> *Clarification on result presentation*
>
> We thank the reviewer for noting the potential ambiguous writing in our presentation of the error bound theorem statements.
> We have rectified the notion of ``error'' (ie. population error $err_\mathcal{D}(h_k)$) in the problem formalization to make these theorems more clear, and have also clarified how the proofs (which bound failure probability to be at most $\delta$) connect to the final theorem result (which bound success probability to be at least $1-\delta$).
>
> *Figure 2 creation?*
>
> We highlight that the second figure is an illustrative example of the bounds, and does not come from any experimental results.
>
> *How might we compare models?*
>
> We first note that our paper provides the first theoretical demonstration and justification which makes such model comparison statically valid in the first place.
> We intend to emphasize this in the final version of our paper with the following expanded intuition.
>
> Practitioners in resource-adaptive learning (or benchmarks) commonly train on different-sized subsets (100,000 vs. 1M samples) and compare the resulting models. However, this practice is merely a heuristic. Standard PAC theory provides a $1-\delta$ guarantee for a single, fixed sample size $k$ and offers no statistical license to compare the learner from $S_k$ to the learner from $S_{k'}$. As noted in our response to reviewer Uvtz, the for all $k$ requirement induces a severe multiple testing problem.
> In proving that the random ordering achieves an optimal statistical rate simultaneously for all prefixes, we prove that this common practice is statistically sound.
> This means a researcher can now, with formal justification, randomly shuffle a benchmark dataset and directly compare the performance of a model trained on the first 100,000 samples against one trained on the first 1M, knowing it is a true apples-to-apples statistical comparison.

---

> > ### Comment · Reviewer_F5au · 2025-11-26
> > **Response to comment by authors**
> >
> > I agree that the theoretical results in this paper points towards the possibility of comparing models with different budgets of training data. But how would we actually compare the performance of a model trained with 100,000 data samples against one trained with 1M samples?
> >
> > The last sentence is a strong claim. When the authors say an apples-to-apples statistical comparison with formal justification, in what sense are we thinking of?

---

### Official Review · Reviewer_A7ii · 2025-11-13

**Soundness:** 4
**Presentation:** 4
**Contribution:** 2
**Rating:** 4
**Confidence:** 3

**Summary:**

The authors introduced the universal ordering problem within the PAC learning framework. As a baseline, they analyzed random orderings using standard techniques. They then refined this analysis via Ville’s inequality, which yields a tighter bound that unexpectedly removes all logarithmic factors.

**Strengths:**

- The paper presents original contributions.
- While I have not verified every proof in full detail, the arguments appear correct.
- I specifically like the Ville’s inequality idea.
- The paper is well-written and easy to follow. I particularly appreciated the step-by-step progression of improvements.
- The paper includes a comprehensive discussion section.

**Weaknesses:**

- The main concern I have is that I’m not convinced of the importance of the problem. In particular, while I understand it on a mathematical level, the stated motivations did not resonate with me.
- The scope of the paper is limited by its exclusive focus on random orderings, which leaves algorithmic aspects unexplored. In a setting where the goal is to minimize the expected error, random ordering is clearly optimal. The work would be substantially more impactful if the high-probability optimal ordering differed from the random ordering.
- The discussion of related works could be strengthened. For example, the following paper appears relevant: https://arxiv.org/pdf/2202.05246

**Questions:**

- Could you elaborate on the key motivations for studying this problem?
- Is there any reason to consider non-random orderings in your formulation?

**Details Of Ethics Concerns:**

-

---

> ### Author Response · Authors · 2025-11-20
> **Response to review**
>
> We thank the reviewer for their positive feedback on our paper's originality, technical correctness, and presentation. We are especially glad that they found the ultimate Martingale's approach with Ville's inequality insightful. We here address some of the specific concerns raised, and encourage the reviewer to read the general comment to all reviewers for more.
>
> *``I'm not convinced of the importance of this problem..."*
>
> We argue that this problem is fundamental to the statistical validity of heuristic machine learning practice.
> Practitioners routinely train models on increasing data subsets to examine scaling laws or compare sample efficiency.
> Standard PAC theory provides no justification for this: while a standard PAC bound holds for fixed $n$, constructing a learning curve and checking the error at various $k \le n$ is a multiple testing problem over correlated datasets.
> Beyond this, in realistic training at scale, jobs are frequently trimmed to compute budget constraints.
> Our work provides the first anytime-valid guarantees for PAC learning, ensuring that if training halts at any arbitrary random point $k$, the resulting model still carries a rigorous generalization guarantee.
>
> *``The scope of the paper is limited by its exclusive focus on random orderings...''*
>
> We highlight that our analysis of a random permutation is not just a simple baseline; it is motivated by the fact that a random ordering is provably the only *task-agnostic* solution to the universal ordering problem that achieves an optimal error rate (see the general comment to all reviewers).
> Specifically, any "clever" or deterministic ordering (e.g., curriculum learning) introduces a specific bias towards the given task.
> An adversary can always construct a PAC-learnable task where this bias is maximally harmful (e.g. by placing all informative samples at the end of the clever ordering).
> This would cause the deterministic ordering to fail the universal "for all $k$" condition.
> A random permutation, by contrast, is a randomized algorithm.
> Its invariance to such adversarial task design provides a robust guarantee that, with high probability, every prefix is a representative random subset.
> Our paper provides the first tight, non-vacuous bounds for this canonical solution.
> Moreover, our main contribution with test martingales surprisingly shows that the simple random ordering does solve this hard problem without any statistical penalty.

---

> > ### Comment · Reviewer_A7ii · 2025-11-27
> >
> > I thank the authors for their response. Your points are well received. I have increased my score. Please incorporate the formal version of your argument regarding the task-agnostic perspective and the underlying motivations you described here into the final version of your paper.

---

### Author Response · Authors · 2025-11-20
**General Rebuttal to All Reviewers**

We thank the reviewers for their insightful comments. We here address the common concern that examining the uniformly random ordering is insufficient.
Specifically, we here showcase that under the \emph{task-agnostic} condition (1) a random ordering must always be better than a deterministic ordering, and (2) the uniformly random ordering achieves the optimal error against an adversary.
We formalize the notion of a task-agnostic ordering via a minimax game: the ordering algorithm must fix its distribution over permutations before the adversary selects the critical samples which reveal the true concept.

**Random vs. Deterministic Ordering:** We first show that a deterministic data ordering is easily exploited and cannot be task-agnostic (and furthermore universally PAC valid).
It is intuitive that any committed to deterministic strategy can be exploited by an adversary that exactly counters this strategy.
We formalize this intuition as follows:

Let $\mathcal{A_det}$ be an algorithm for deterministic ordering of a dataset.
We proceed to construct a task which has maximal loss for the algorithm.
Let $\mathcal{H_adv} = \{h_0, h_1\}$ where $h_0(x) = 0$ and $h_1(x) = 1$ for all $x \in \mathcal{X}$ (a simple PAC-learnable hypothesis class with \texttt{VC} dimension 1).
Furthermore let $\mathcal{X} = \{x_A, x_B\}$ where $x_A$ has label 0 and $x_B$ has label 1.
Assume without loss of generality that $\mathcal{A_det}$ sorts all instances of $x_A$ before $x_B$.
We then construct the adversarial task such that $h_1$ is the true hypothesis but is hard to discover: for $\epsilon \in (0, \frac12)$ define the distribution $\mathcal{D_adv}$ by $\mathbf{Pr}(x_B) = 1-\epsilon$ and $\mathbf{Pr}(x_A) = \epsilon$.
Note that the true hypothesis would predict 1 for $x_B$ and 0 for $x_A$, but within the hypothesis class $\mathcal{H_adv}$ the only hypothesis with error at most $\epsilon$ is $h_1$.
The error of $h_0 = 1-\epsilon$.
Now, assume we have a consistent learner who draws an $n$ sample set from $\mathcal{D_adv}$.
Let $E$ be the event that this sample set contains at least one $x_A$ sample: $\mathbf{Pr}[E] = 1 - (1-\epsilon)^n$ which tends to 1 as $n \rightarrow \infty$.
Conditional on $E$, we have that the sampled set contains $k \ge 1$ instances of $x_A$ and $n-k$ of $x_B$.
The algorithm applied to this set produces an ordering where the first $k$ elements are $x_A$.
For the prefixes up to $k+1$, the learner only sees samples with label 0 and returns hypothesis $h_0$.
Therefore, the deterministic algorithm is easily exploited to achieve maximal error on the prefix and we have the result (note that we can reverse the task / algorithm wlog).

**Optimality of Uniformly Random:** Further proving that the uniform random ordering is the unique optimal strategy relies on a similar construction and intuition.

Let $S_k$ be the set of indices in the $k$-sized prefix of a dataset.
For any randomized ordering algorithm (which forms a distribution over the space of permutations $\mathcal{P}$), the expected number of elements in $S_k$ must be exactly $k$.
By linearity of expectations, we have
$$\sum_{i=1}^n \mathbf{Pr}_{\pi \sim \mathcal{P}}[i \in S_k] = \mathbf{Ex}[|S_k|] = k$$
which implies that the average inclusion probability for any data point in the set of size $k$ is $k/n$, regardless of the algorithm.
Similar to above, we consider the adversary who places a critical sample, $x^*$, (which fully reveals the hypothesis) at the index $i$ which minimizes the inclusion probability. The worst-case success probability is then given by: $\arg\min_i \mathbf{Pr}[i \in S_k]$.
Moreover, because the average is fixed at $k/n$, this minimum is maximized if and only if all such probabilities are equal
$$\min_i \mathbf{Pr}[i \in S_k] \le \frac{k}{n}$$
Equality holds iff the algorithm assigns equal marginal probability to every $i \in [n]$.
Since any non-uniform distribution must have some index $j$ such that $\mathbf{Pr}[j \in S_k] < k/n$, the adversary can exploit and increase the failure rate.
Thus, the uniformly random ordering is the unique best strategy.

**We intend to include formalized versions of these results in the final version of our paper to better motivate the study. We thank the reviewers for their concerns as they lead us to these results!**

---

> ### Comment · Reviewer_F5au · 2025-11-26
> **Response to general rebuttal**
>
> In the first proof, if there exists data with both labels $0, 1$ and your hypothesis class only contains two constant functions, doesn't that rule out the existence of a consistent learner?
>
> I believe the authors can revise their paper for ICLR during the discussion period. Would you be able to include these results in an updated version within this period?

---

### Meta-Review · Area_Chair_ewLg · 2025-12-19

**Summary:**

The paper introduces a new "universal ordering" problem for PAC learning. The goal is to order a set of n points such that for every k the first k points are a near-optimal subset of k points for training a PAC learning.

The paper considers a random permutation of the given n points, and shows that for all k, a random permutation achieves no overhead compared to a random subset of size k. This is a good result.

I think this is a good paper, but it is difficult to recommend acceptance at this point. Reviewers have raised serious concerns regarding writing. Moreover, the new result regarding task-agnostic optimality of random permutations strengthens the paper but probably also demands a fresh review since it is a significant addition. Therefore, the suitable course appears to be to have the paper be resubmitted.

**Reviewer Concerns:**

The main concerns are regarding the focus on random orderings, and the writing.

The authors justify the focus on random orderings in the discussion period by showing a new result. They showed that random orderings are optimal in a minimax game where the ordering algorithm must fix its distribution over permutations before the adversary selects the critical samples which reveal the true concept. I think this result strengthens the message of the paper.

The other concern is regarding writing, raised by reviewers RSnn and 5LRS who both gave a score of 2. It is not clear if this is adequately addressed at this point.

**Reviewer Scores:**

One of the reviewers (A7ii) said they will increase their score. It is quite possible that one or two others may increase as well, since nearly all raised the concern about the focus on random orderings.

---

### Decision · Program_Chairs · 2026-01-26

Reject